# Reliability Analysis of Nuclear Power Plant Electrical System Considering Common Cause Failure Based on GO-FLOW

**Zhijian Wang [1], Yao Sun [1], Jie Zhao [2,\*], Xuzhu Dong [2], Chen Chen [1], Bo Wang [2] and Haocheng Wu [2]**

[1] China Nuclear Power Engineering Co., Ltd., Haidian District, Beijing 100840, China; wangzj@cnpe.cc (Z.W.); sunyao@cnpe.cc (Y.S.); chenchenb@cnpe.cc (C.C.)

[2] Hubei Engineering and Technology Research Center for AC/DC Intelligent Distribution Network, School of Electrical Engineering and Automation, Wuhan University, Wuhan 430072, China; dongxz@whu.edu.cn (X.D.); whwdwb@whu.edu.cn (B.W.); qczedae1956355@163.com (H.W.)

\* Correspondence: jiez_whu@whu.edu.cn

**Abstract:** The reliability of nuclear power plant electrical systems is an important guarantee of nuclear safety, and the common fault failure problem arising from redundant design and intelligent control may greatly affect reliability assessment results. Combined with the features of repairability, multi-state characteristics, and common fault failure of nuclear power plant electrical systems, a reliability analysis method of nuclear power plant electrical systems based on the GO-FLOW method considering common fault failure is proposed. This study firstly constructs the algorithmic model of combining operators of repairable components and the equivalent model of reliability parameters of multi-mode repairable components, then establishes a probability calculation model of common fault failure for repairable systems by considering the quantitative computation of the common signaling system model, and finally, quantitatively calculates the reliability of nuclear power plant electrical systems and their influencing factors. The example simulation calculates the reliability of the external power supply system and the electrical system of the nuclear power plant, analyzes the influence of the common signal processing and the common fault failure factors on the reliability of the electrical system of the nuclear power plant, and verifies the validity of the proposed method. The results show that the common fault failure factors have a large impact on the system reliability analysis; the common fault failure of the standby diesel generator set will seriously reduce the reliability of the electrical system, which can be improved by installing additional standby diesel generators.

**Keywords:** nuclear power plant electrical systems; reliability; GO-FLOW; common fault failure

## 1. Introduction

The safety of nuclear power has significant political and social implications, and nuclear accidents, once they occur, will affect social stability, and cause incalculable losses to the safety of people's lives and property. The Three Mile Island nuclear accident in 1979 and the Chernobyl accident in 1986 in the former Soviet Union have mercilessly reminded people that the safety of nuclear power is still the most important research topic in the process of nuclear power development. In March 2011, earthquakes and tsunamis caused a series of serious accidents, such as the failure of the power supply outside the Fukushima nuclear power plant in Japan and the failure of the back-up power supply inside the plant, and ultimately triggered a series of serious accidents, such as the leakage of radiation, which had a far-reaching impact on the world. This has had far-reaching impacts worldwide. This shows that safety is always an important factor affecting the development of nuclear power.

Normal operation of the electrical system is a prerequisite for the effective play of nuclear safety defense, which plays an important role in nuclear power plant shutdown, waste heat export, emission control, etc. It is also an important guarantee for the safety of the conventional island of the nuclear power plant and is also the basis for many types of

control equipment to work reliably. The reliability analysis of nuclear power plant electrical systems has important theoretical value and practical significance [1,2].

At present, there are mainly these commonly used methods for reliability analysis of nuclear power plant electrical systems: (1) The fault tree analysis method, which consists of specific logic gates and various fault events, has a simple structure, and is easy to analyze. However, the modeling is greatly influenced by human factors, and it is easy to miss or repeat. (2) The Monte Carlo method, with which it is easy to model complex systems, and the model can take time constraints, parameter distributions, etc., into account. However, it requires a lot of operations, which is expensive and time-consuming. (3) The Markov state transfer method, which can explicitly consider the effects of changes in process variables on the system state and the system control rules, as well as the effects of time changes on the system state. However, for large and complex systems, as the number of components in the system increases, the states in the Markov model will increase exponentially, and it is more difficult to use the Markov state transfer model alone for dynamic probabilistic evaluation. (4) The GO analysis method, which can directly simulate the interactions and correlations of the system and components, and directly perform the system success probability analysis in a success-oriented manner. However, there are many types of operators, and the use of them is complicated, which requires analysts not only to be very familiar with the system but also to have an in-depth understanding of the GO method. In addition, the GO method program development is more complicated, and there is no complete calculation program yet. (5) The GO-FLOW analysis method, with which it is easy to model, change, and verify. A GO-FLOW diagram can fully express the relationship and interaction between components and systems and can well describe and reflect the system structure.

Probabilistic safety evaluation is a common evaluation method in nuclear power plants, which can be categorized into static methods and dynamic methods. Static methods include reliability block diagram analysis, methods based on binary decision diagrams, etc. [3–6]. Dynamic methods include continuous event trees, event sequence diagrams, the GO-FLOW method, etc. [7–11]. The GO-FLOW method is a new probabilistic risk evaluation method. It uses a graphical rendition of success as a guide to the system and can assess the reliability and availability of the system. The specific process of using the GO-FLOW method for system reliability analysis is as follows: (1) Analyze the given system; determine the scope, function, and included components of the system; specify the system's reliability indexes; and give an engineering or schematic diagram of the system. (2) Determine the inputs and outputs of the system. (3) Determine the minimum set of output signals required for normal operation of the system. (4) Identify the system's reliability indicators. (5) Specify the normal operating state of the system and determine the minimum set of output signals required for normal operation of the system. (6) Create a GO-FLOW diagram and determine the type of GO-FLOW operator to be used according to the function of the unit; the operator should contain all the major units in the system. (7) Define a finite number of discrete time points. (8) Once the GO-FLOW diagram is established, determine the state probability data for all units and then enter the data by operator number. (9) Starting from the input operators, compute the output signals of the system step-by-step up to the output signals of the system based on the GO-FLOW diagram and the data according to the rules of operation of the operators. (10) Based on the results of the GO-FLOW operation and the system success criteria used to calculate the reliability or availability of the system, evaluate the system according to the functions and requirements of the system. Nuclear power plant systems are becoming more and more complex, and traditional probabilistic safety assessments for nuclear power plants analysis techniques have great limitations. It has become an inevitable trend to use dynamic safety analysis techniques for system evaluation. For example, reference [12] used the GO-FLOW method to study the aging problem of nuclear power plants. Reference [13] used a dynamic event tree and the GO-FLOW method for relay delay system reliability analysis. The GO-FLOW method is suitable for solving the time correlation problem of dynamic system safety analysis [14]. Directly applying the traditional GO-FLOW method for reliability

analysis of nuclear power plant electrical systems has the following problems: (1) In order to be able to use the GO-FLOW method for nuclear power plant electrical systems, its basic operators must be improved so that the variables are successfully introduced into the operation rules, including regarding component repair rate [15]. In addition, the improved operator model can be used to simulate the equipment and components of a nuclear power plant electrical system and combined with the equivalent reliability parameters of the multi-modal repairable components to analyze the reliability of a nuclear power plant electrical system [16]. (2) The structure of the electrical system of a nuclear power plant is complex, and if it is calculated directly according to the arithmetic rules of the operator, the results will have errors, which will affect the accuracy of the system assessment [17,18]. (3) Common Fault Failure (CFF) is the simultaneous failure of several units in a system due to some common reasons (e.g., changes in the environment, events such as earthquakes or lightning strikes, operational errors in maintenance, other human interference, etc.). Since the unit normal or fault states are statistically related to each other in a system with common fault failure, the reliability data processing and analyzing methods of the system are quite different from the traditional ones, which brings more difficulties to the reliability analysis of the system [17,18]. Probabilistic risk analysis in the nuclear industry shows that common fault failure is one of the main reasons for system failure, equipment unavailability, and risk in nuclear power plants [19,20]. In the current literature, few scholars have used the GO-FLOW method to analyze the impact of common cause failure factors on system reliability. In this paper, combined with the characteristics of a repairable nuclear power plant power supply system, a general method of using the GO-FLOW method to analyze the influence of common cause failure factors on the reliability of repairable systems is studied.

In this paper, the algorithmic model of combining operators for repairable components and the equivalent model of reliability parameters for multi-modal repairable components are constructed, the GO-FLOW model of the electrical system of a nuclear power plant is established by considering the improved quantitative computation method of the common signal system, and the probabilistic computation model of the co-causal failure taking into account repairable features of the electrical system of a nuclear power plant and the repairable system under the influence of co-causal failure factors based on the GO-FLOW method is proposed. The calculation process of the repairable system under the influence of common failure factors based on the GO-FLOW method is presented. The example calculation analyzes the influence of single power supply failure and common fault failure factors on the reliability of nuclear power plant electrical systems, verifies the validity of the proposed method, and proposes measures such as adding backup diesel generators to improve the reliability of nuclear power plant electrical systems.

The main objectives of this paper's work are (1) to establish a reliability model of nuclear power plant electrical systems based on the GO-FLOW method, (2) to study the analysis method of repairable systems' common fault failure based on the GO-FLOW method, (3) to analyze the process of reliability analysis of nuclear power plant electrical systems based on the GO-FLOW method taking into account the common fault failure, and (4) to carry out the arithmetic case analysis so as to draw the conclusions.

## 2. Reliability Modeling of Nuclear Power Plant Electrical Systems Based on the GO-FLOW Method

The electrical system of a nuclear power plant includes the off-plant electrical system and the on-plant electrical system, which mainly consists of the main turbine generator, the main external power supply, the auxiliary external power supply, the emergency diesel generator, the main transformer, the plant transformer, the auxiliary transformer, the main wiring, the busbar, the generator outlet circuit breaker, the circuit breaker, and the disconnecting switch, as shown in Figure 1. Among them, the off-plant electrical system includes the main external power supply and auxiliary external power supply, and the on-plant electrical system includes the main turbine generator and diesel generator set. The total installed capacity of nuclear power plants is 6.12 million kilowatts (MW), and

large-scale nuclear power plants usually have a single unit capacity of 900 MW or 1300 MW, a transmission level of 400 kV or 500 kV, and megawatt-class pressurized water reactors. When the main turbine generator fails, the generator outlet circuit breaker trips, and the plant AC power is provided by the extra high-voltage main external power supply. When both the generator power supply and the extra high-pressure main external power supply are lost at the same time, the plant AC power supply is switched from the external standby power supply through the fast-cutting device. When the above situation occurs, the two standby diesel generator sets in the plant start to supply power to the emergency auxiliary equipment of the nuclear power plant.

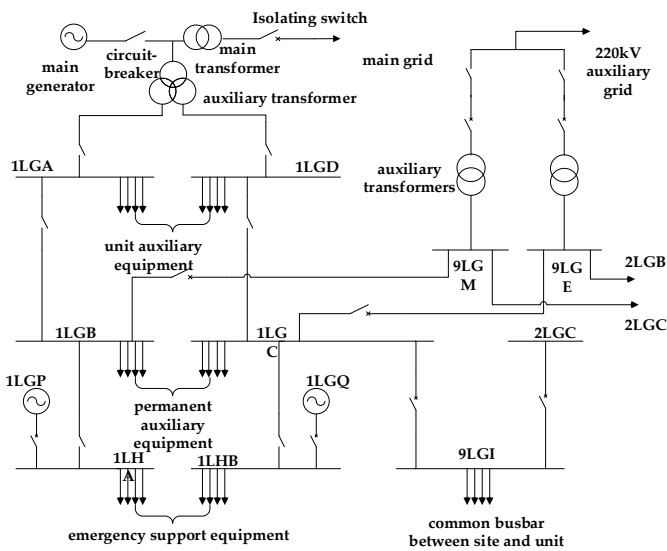

**Figure 1.** Schematic diagram of nuclear power plant electrical system.

The reliability analysis task is to obtain the trend of system reliability over time. When applying the GO-FLOW method for electrical system reliability analysis [21], the system analysis is carried out first, and the GO-FLOW diagram of the electrical system of the nuclear power plant established is shown in Figure 2. In this case, type 21 and 35 operators are employed in series to represent a repairable working component, and the type 35 operator simulates a working component that fails over time. Type 26 and 37 operators are employed in series to represent a repairable conduction element, and the type 37 operator simulates a conduction element that fails over time. By setting a series of time points in the event sequence, completing the state setting of the type 25 operator at each time point, inputting the reliability parameters of each repairable component, and performing the GO-FLOW operation, the trend of the electrical system success probability and failure probability over time can be directly obtained. In this case, the reliability assessment mainly involves the external power supply system and the whole plant's electrical system.

(1) Failure analysis of the external power supply system: In the GO-FLOW diagram of the external power supply system, signal flow 54 represents the entire external power supply system, and signal flows 34 and 53 represent the outputs of the two-way power supply of the external power supply system, respectively. The logical relationship of the two-way power supply outputs is indicated by OR gate operator 54, and when there is no power supply in the two-way power supply because of a loss of the external power supply system, this is called a LOOP event.

(2) Analysis of the failure of the electrical system and plant-wide power failure: When the external power system of a nuclear power plant is lost, the external power system and the diesel generator relate to the contingency gate operators 55 and 59 in the diagram. Signal flow 58 and 62, respectively, represent the backup system two-way power supply output, with OR gate operator 63 representing the two-way power

supply output of the logical relationship. The output signal 63 failure means that the two-way power supplies have no power supply output because of the nuclear power plant's plant-wide power outage. This is called an SBO event.

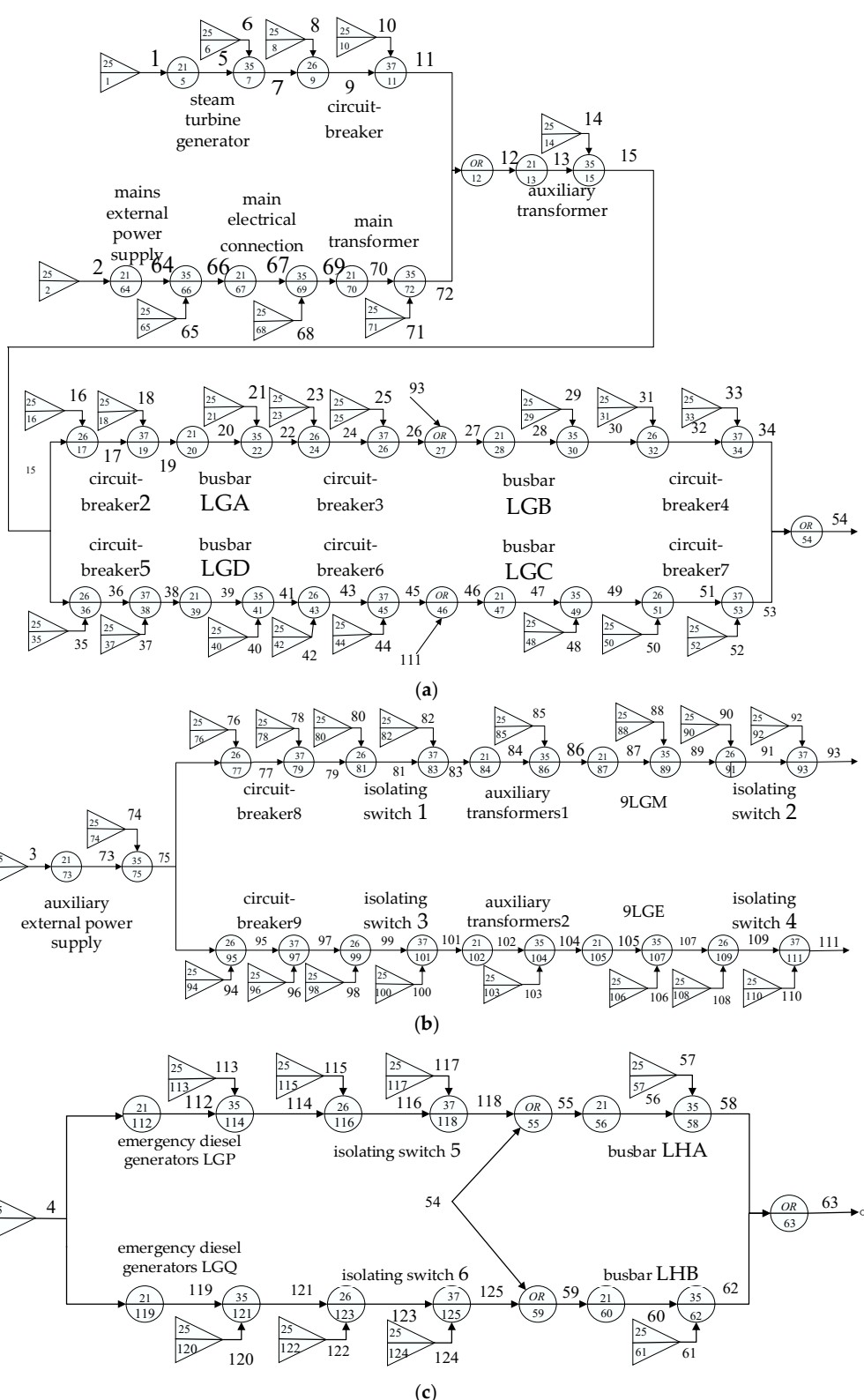

**Figure 2.** GO-FLOW diagram of the electrical system of a nuclear power plant. (**a**) Main power system. (**b**) Auxiliary power systems. (**c**) Backup power systems.

### 3. A Common Fault Failure Analysis Method for Repairable Systems Based on the GO-FLOW Approach

*3.1. Theoretical Basis of GO-FLOW Method to Deal with Common Fault Failure*

The GO-FLOW method avoids many complex calculations brought about by the simultaneous treatment of multiple co-causal failures of various system components and solves the problem of the fault tree method being unable to realize the application of reliability analysis in co-causal failure systems due to the generation of many cut-sets [22–24].

By systematic argument [21], for two units *A* and *B*, the system failure rate *F (A, B)* is as follows:

$$F(A, B) = F_0 + [F(1,1) - F(0,0)] \bullet C_{A,B} \tag{1}$$

where $F_0$ is the probability of system failure without considering the common fault failure, $F(0,0)$ is the probability of system failure when the probability of failure of unit *A* and unit *B* are both taken as 0, $F(1,1)$ is the probability of system failure when the probability of failure of unit *A* and unit *B* are both taken as 1, and the probability of common fault failure of unit *A* and unit *B* is $C_{A,B}$.

For more complex systems where there exists a co-causal failure unit group, the co-causal failure system failure rate $F_k$ is as follows:

$$F_k = F_0 + \sum_{j=1}^{k} C_j [F(1,1,\cdots) - F(0,0,\cdots)] \tag{2}$$

where $F_0$ is the probability of system failure without considering common fault failure, $C_j$ is the probability of common fault failure for group *j*, $F(1,1,\cdots)$ is the probability of system failure when all common cause unit failure probabilities are taken as 1, and $F(0,0,\cdots)$ is the probability of system failure when all common cause unit failure probabilities are taken as 0.

Equation (2) shows that the total system unreliability consists of two components: ① the system unreliability caused by the independent failure of each component, i.e., $F_0$, and ② system unreliability caused by various common fault failure events.

The following are examples of two sets of common fault failure components: (1) Auxiliary transformers 1 and 2 are designed and manufactured by the same manufacturer and have the potential for common fault failure. (2) Diesel generators are usually placed in the same engine room with the same environment and are subject to common fault failure during startup and operation.

*3.2. Model for Calculating the Probability of Co-Causal Failure of Repairable Systems*

Determining the probability of occurrence of a common fault failure is the key to solving Equation (2). For the component group that does not consider maintenance, the probability of common fault failure is calculated by the common fault failure parameter model, and there are existing common fault failure parameter models, such as the α-factor model, β-factor model, and so on. Given the estimated values of the model parameters, it is possible to calculate the probability of common fault failure for a group of components without considering maintenance. The probability of common fault failure for the group of repairable components is calculated using the following method.

Repairable components in engineering are usually assumed to follow an exponential distribution, and the common fault failure rate is obtained from the common fault failure parameter method. Consider that two repairable components *A* and *B* have a common fault failure with a common cause failure rate of *c*. The failure rate of component *A* is $\lambda_A = \lambda_1 + c$ and the repair rate is $\mu_1$; the failure rate of component *B* is $\lambda_B = \lambda_2 + c$ and the repair rate is $\mu_2$.

The group of two repairable parts with common fault failure has a total of five states:

State 0: Parts *A* and *B* are intact,
State 1: Component *A* has failed, and component *B* is intact,

State 2: Component *A* is intact, and component *B* has failed,
State 3: Non-co-causal simultaneous failure of components *A* and *B*,
State 4: Co-causal failure of components *A* and *B*.

For the above case, a Markov process can be used. A Markov process, also called a Markov chain, is a memoryless stochastic process; it can be represented by a tuple <*S*, *P*>, where *S* is a finite number of state sets and *P* is the state transfer probability matrix. The application of the Markov process gives the state transfer diagram shown in Figure 3.

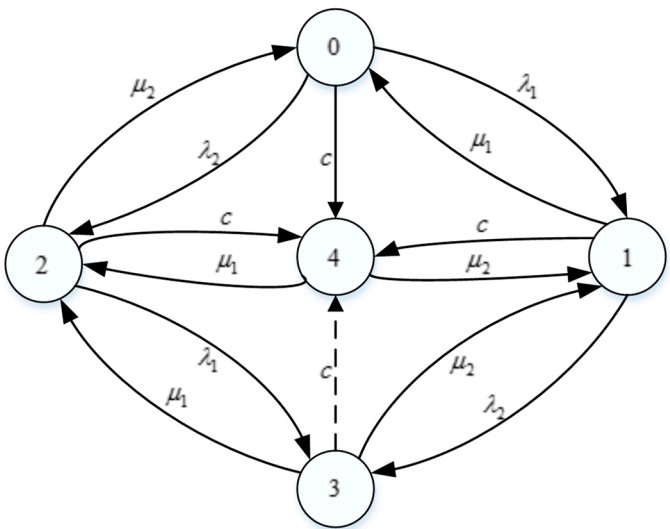

**Figure 3.** State transfer diagram of the system with two repairable components.

Observing the Markov state transfer diagram, it can be concluded that a dotted line from state 3 to state 4 is added to the state transfer diagram. At this point, considering states 0, 1, 2, and 3 as total states without common fault failure and state 4 as a state where common fault failure occurs, the approximate formula for the probability of common fault failure with time *t* can be derived as:

$$C_{A,B}(t) = P_4(t) = \frac{c}{c + \mu_1 + \mu_2} + \left[ \gamma_c - \frac{c}{c + \mu_1 + \mu_2} \right] e^{-(c+\mu_1+\mu_2)t} \tag{3}$$

where *c* is the approximate equivalent failure rate of the two components with common fault failure, $\mu_1 + \mu_2$ is the repair rate, and $\gamma_c$ is the probability that the two components are in a state of common fault failure.

For more complex systems with *n* repairable components, the approximation of the probability of common fault failure at time *t* is as follows:

$$C_n(t) = \frac{c}{c + \sum\limits_{i=1}^{n} \mu_i} + \left[ \gamma_c - \frac{c}{c + \sum\limits_{i=1}^{n} \mu_i} \right] \exp\left[ -\left( c + \sum\limits_{i=1}^{n} \mu_i \right) t \right] \tag{4}$$

where *c* is the approximate equivalent failure rate of co-causal failure of *n* components, $\sum\limits_{i=1}^{n} u_i$ is the repair rate of repairable components, and $\gamma_c$ is the probability that two components are in a state of co-causal failure.

Equations (3) and (4) give simple approximate expressions for calculating the probability of common fault failure from the common fault failure rate, and it is then possible to use Equations (1) and (2) to perform state probability calculations for repairable systems containing common fault failure.

## 4. Reliability Analysis Process of Nuclear Power Plant Electrical Systems Based on GO-FLOW Method Considering Common Fault Failure

The quantitative computational treatment of the GO-FLOW method in a co-causal failure system is utilized, and its computational flow is shown in Figure 4.

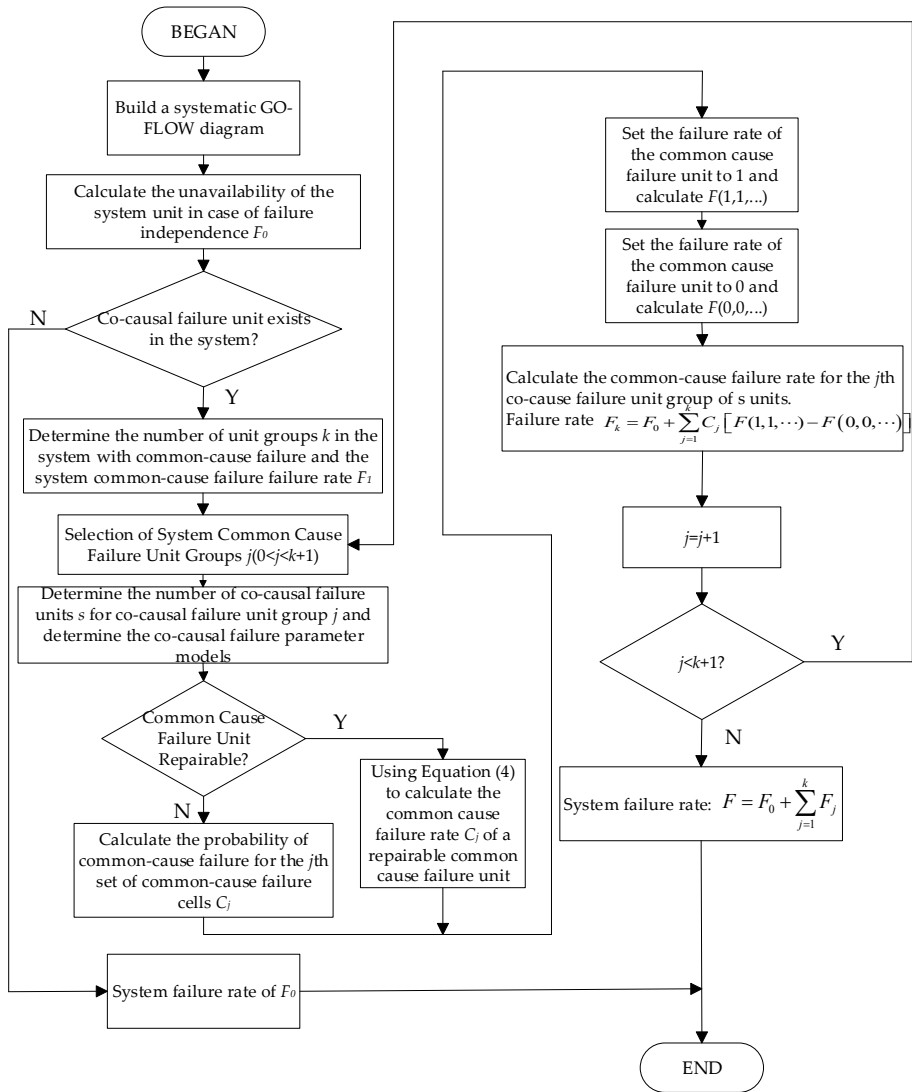

**Figure 4.** Reliability analysis flow of nuclear power plant electrical system based on GO-FLOW method considering common fault failure.

The main steps include, based on the establishment of the system GO-FLOW diagram, calculating the system unreliability caused by the independent failure of each component $F_0$. If the system does not have a common fault failure unit, then the system failure rate is $F_0$; if the system has a common fault failure unit, it is necessary to determine the number of unit groups in the system that have a common fault failure, $k$, the system failure rate of the common fault failure $F_1$, the number of units in the common fault failure units of the unit group $j$ and $s$, and the common fault failure of the parameter model. If the common fault failure unit can be repaired, the common fault failure rate $C_j$ of the repairable common fault failure unit is calculated according to Equation (4); if it cannot be repaired, the common fault failure probability $C_j$ of the $j$th group of the common fault failure units is calculated. The failure probability of the common failure unit is set to 1 and 0, and the failure probability of the common fault failure unit is calculated at $F(1, 1, \cdots)$ and $F(0, 0, \cdots)$, respectively, and at this time, the common fault failure rates of the s units in the $j$th group of common fault failure units can be calculated by Equation (2); each loop

increases *j* by 1; when *j* is greater than *k* + 1, the output system failure rate is *F*; otherwise, the calculation operation is repeated.

## 5. Example Analysis

### 5.1. Calculations Introduction

Taking the electrical system of a nuclear power plant shown in Figure 1 as an example, the probability of losing the main external power supply is taken as $2.3 \times 10^{-3}$ times/machine·years, and the average repair time is 100 h. The failure frequency and average repair time of some repairable components are shown in Table 1, and the overhaul frequency and overhaul time are shown in Table 2. Among them, the isolation switches in the system are considered to be non-failure parts, and the success probability is always considered to be 1. The equivalent failure rate, equivalent repair rate, average success probability, and average failure probability of each part can be found out through the comprehensive processing, respectively (Table 3).

**Table 1.** Frequency of failures and average repair time for repairable components.

| Unit Name | Malfunctions Number of Modes | Failure Frequency (Times/Year) | Average Repair Time (Hours) |
|---|---|---|---|
| main steam turbine generator | 1 | 1.500 | $7.300 \times 10$ |
| mains external power supply | 1 | $2.300 \times 10^{-3}$ | $1.000 \times 10^2$ |
| auxiliary external power supply | 1 | 8.468 | 3.351 |
| emergency diesel generator | 1 | $1.743 \times 10^2$ | 5.000 |
| main transformer | 2 | $2.015 \times 10^{-3}$ $8.760 \times 10^{-3}$ | $1.000 \times 10$ $7.500 \times 10^2$ |
| auxiliary transformer | 2 | $5.694 \times 10^{-3}$ $8.760 \times 10^{-3}$ | $1.000 \times 10$ $4.000 \times 10^2$ |
| auxiliary transformers | 2 | $1.139 \times 10^{-2}$ $1.139 \times 10^{-2}$ | $1.000 \times 10$ $4.000 \times 10^2$ |
| main electrical connection | 1 | $2.059 \times 10^{-2}$ | $1.060 \times 10^2$ |
| busbar | 2 | $3.679 \times 10^{-3}$ $4.643 \times 10^{-4}$ | $5.000 \times 10$ $7.200 \times 10$ |
| generator export circuit-breaker | 1 | $4.890 \times 10^{-2}$ | $1.116 \times 10^2$ |
| circuit-breaker | 1 | $3.030 \times 10^{-2}$ | $5.750 \times 10$ |
| isolating switch | 1 | 0 | 0 |

**Table 2.** Frequency and duration of overhaul of repairable parts.

| Unit Name | Unit State | Frequency of Overhaul (Times/Year) | Overhaul Time (Hours) |
|---|---|---|---|
| main steam turbine generator | examine and fix (a motor) | 1.000 | $2.800 \times 10^2$ |
| main transformer | examine and fix (a motor) | $2.000 \times 10^{-1}$ | $1.600 \times 10^2$ |
| main electrical connection | examine and fix (a motor) | $5.000 \times 10^{-1}$ | $2.400 \times 10$ |
| circuit-breaker | examine and fix (a motor) | $7.872 \times 10^{-1}$ | $8.100 \times 10$ |

**Table 3.** Reliability data for repairable components of this nuclear power plant.

| Unit Name | Equivalent Failure Rate | Equivalent Maintenance Rate | Average Probability of Success | Mean Failure Probability |
|---|---|---|---|---|
| main steam turbine generator | 2.50000 | $5.62260 \times 10$ | 0.957429 | $4.2571 \times 10^{-3}$ |
| circuit-breaker 1 | $4.89000 \times 10^{-2}$ | $7.84946 \times 10$ | 0.999377 | $6.2258 \times 10^{-4}$ |
| auxiliary transformer | $1.44540 \times 10^{-2}$ | $3.55572 \times 10$ | 0.999594 | $4.0633 \times 10^{-4}$ |
| circuit-breaker 2 | $8.17500 \times 10^{-1}$ | $1.09430 \times 10^{2}$ | 0.992585 | $7.4151 \times 10^{-3}$ |
| busbar LGA | $4.14348 \times 10^{-3}$ | $7.00386 \times 10^{2}$ | 0.999994 | $5.9160 \times 10^{-6}$ |
| circuit-breaker 3 | $8.17500 \times 10^{-1}$ | $1.09430 \times 10^{2}$ | 0.992585 | $7.4151 \times 10^{-3}$ |
| busbar LGB | $4.14348 \times 10^{-3}$ | $7.00386 \times 10^{2}$ | 0.999994 | $5.9160 \times 10^{-6}$ |
| circuit-breaker 4 | $8.17500 \times 10^{-1}$ | $1.09430 \times 10^{2}$ | 0.992585 | $7.4151 \times 10^{-3}$ |
| circuit-breaker 5 | $8.17500 \times 10^{-1}$ | $1.09430 \times 10^{2}$ | 0.992585 | $7.4151 \times 10^{-3}$ |
| busbar LGD | $4.14348 \times 10^{-3}$ | $7.00386 \times 10^{2}$ | 0.999994 | $5.9160 \times 10^{-6}$ |
| circuit-breaker 6 | $8.17500 \times 10^{-1}$ | $1.09430 \times 10^{2}$ | 0.992585 | $7.4151 \times 10^{-3}$ |
| busbar LGC | $4.14348 \times 10^{-3}$ | $7.00386 \times 10^{2}$ | 0.999994 | $5.9160 \times 10^{-6}$ |
| circuit-breaker 7 | $8.17500 \times 10^{-1}$ | $1.09430 \times 10^{2}$ | 0.992585 | $7.4151 \times 10^{-3}$ |
| busbar LHA | $4.14348 \times 10^{-3}$ | $7.00386 \times 10^{2}$ | 0.999994 | $5.9160 \times 10^{-6}$ |
| busbar LHB | $4.14348 \times 10^{-3}$ | $7.00386 \times 10^{2}$ | 0.999994 | $5.9160 \times 10^{-6}$ |
| mains external power supply | $2.30000 \times 10^{-3}$ | $8.76000 \times 10$ | 0.999974 | $2.6255 \times 10^{-5}$ |
| main electrical connection | $7.50000 \times 10^{-1}$ | $4.10625 \times 10^{2}$ | 0.998177 | $1.8232 \times 10^{-3}$ |
| main transformer | $2.10775 \times 10^{-1}$ | $4.78461 \times 10$ | 0.995614 | $4.3860 \times 10^{-3}$ |
| auxiliary transformer | 8.46800 | $2.61414 \times 10^{3}$ | 0.996771 | $3.2288 \times 10^{-3}$ |
| circuit-breaker 8 | $8.17500 \times 10^{-1}$ | $1.09430 \times 10^{2}$ | 0.992585 | $7.4151 \times 10^{-3}$ |
| isolating switch 1 | 0 | $+\infty$ | 1 | 0 |
| auxiliary transformers 1 | $2.27760 \times 10^{-2}$ | $4.27316 \times 10$ | 0.999467 | $5.3281 \times 10^{-4}$ |
| 9LGM | $4.14348 \times 10^{-3}$ | $7.00386 \times 10^{2}$ | 0.999994 | $5.9160 \times 10^{-6}$ |
| isolating switch 2 | 0 | $+\infty$ | 1 | 0 |
| circuit-breaker 9 | $8.17500 \times 10^{-1}$ | $1.09430 \times 10^{2}$ | 0.992585 | $7.4151 \times 10^{-3}$ |
| isolating switch 3 | 0 | $+\infty$ | 1 | 0 |
| auxiliary transformers 2 | $2.27760 \times 10^{-2}$ | $4.27316 \times 10$ | 0.999467 | $5.3281 \times 10^{-4}$ |
| 9LGE | $4.14348 \times 10^{-3}$ | $7.00386 \times 10^{2}$ | 0.999994 | $5.9160 \times 10^{-6}$ |
| isolating switch 4 | 0 | $+\infty$ | 1 | 0 |
| emergency diesel generator LGP | $1.74324 \times 10^{2}$ | $1.75200 \times 10^{3}$ | 0.909504 | $9.0496 \times 10^{-2}$ |
| isolating switch 5 | 0 | $+\infty$ | 1 | 0 |
| emergency diesel generator LGQ | $1.74324 \times 10^{2}$ | $1.75200 \times 10^{3}$ | 0.909504 | $9.0496 \times 10^{-2}$ |
| isolating switch 6 | 0 | $+\infty$ | 1 | 0 |

*5.2. Reliability Analysis of Nuclear Power Plants' External Power Supply System and Electrical System*

The probability of success and the probability of failure are calculated at each point in time during the startup and operation phases, as shown in Figure 5, which sets up four scenes:

(1)　Scene 1: The external power supply system in the case of shared signals.
(2)　Scene 2: The external power system when shared signals are not considered.
(3)　Scene 3: The electrical system not considering shared signals.
(4)　Scene 4: The electrical system considering shared signals.

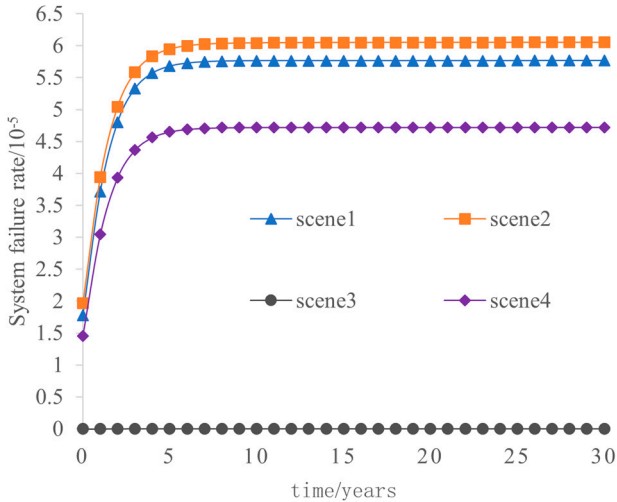

**Figure 5.** Nuclear power plant external power supply and electrical system unreliability.

As can be seen in Figure 5:

(1) For a nuclear power plant's external power supply system, the system failure probability without considering the shared signals grows rapidly from $1.777503 \times 10^{-5}$ at time point 0 to $5.760142 \times 10^{-5}$ at time point 9, and is finally maintained at a relatively smooth state. The system failure rate considering the shared signal grows rapidly from $1.966868 \times 10^{-5}$ at time point 0 to $6.048316 \times 10^{-5}$ at time point 16, and is finally maintained at a relatively smooth state. For the nuclear power plant power supply system, the system failure probability without considering the shared signals is basically maintained near 0. The system failure rate considering the shared signals grows rapidly from $1.456204 \times 10^{-5}$ at time point 0 to $4.721015 \times 10^{-5}$ at time point 14, and is finally maintained at a relatively smooth state. In the early stage of the operation of the nuclear power plant, the failure probability of the external power system and the electrical system increases significantly with time, which is mainly caused by the increase in the failure probability of the equipment and components in the nuclear power plant as well as the connection between the nuclear power plant and the external power grid over time. However, after the 6th year of operation, the probability of failure of the external power system tends to level off, which is mainly caused by the high maintenance rate and regular overhaul of the equipment and components in the nuclear power plant as well as the connection between the nuclear power plant and the external power grid. Overall, the reliability of the external power supply system and the electrical system of the nuclear power plant is quite high.

(2) Compared with the external power supply system, the electrical system has also added two emergency diesel generators to supply power to the emergency bus, resulting in the reliability of the electrical system of the nuclear power plant being two orders of magnitude higher than that of the external power supply system, and the data are in line with the design logic of gradual enhancement of the mitigation measures from the LOOP event to the SBO event.

(3) The results after considering the shared signals are very different from the results without considering the shared signals. Therefore, it is necessary to consider the effect of shared signals for external power systems and electrical systems in nuclear power plants, as well as in redundant systems where special treatment of shared signals is necessary.

*5.3. Analysis of the Impact of Single Power Supply Failure on the Reliability of Electrical Systems*

To explore the importance of the four types of power supply in nuclear power plants, these four types of power supply are sequentially removed from the system to obtain

the change in the reliability of the electrical system of a nuclear power plant under four scenarios, which is shown in Figure 6.

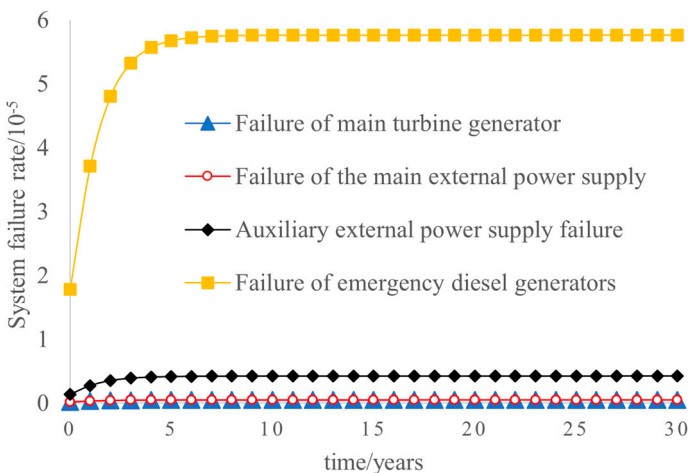

**Figure 6.** Variation curve of unreliability of nuclear power plant electrical system.

From Figure 6, the failure probability of the main turbine failure and the failure probability of the main external power supply failure basically remain near zero. The failure probability of auxiliary external power supply failure grows slowly from $0.136251 \times 10^{-5}$ at time point 0 to $0.423153 \times 10^{-5}$ at time point 10, and is finally maintained at a relatively stable state. The failure probability of emergency diesel generator failure grows rapidly from $1.777507 \times 10^{-5}$ at time point 0 to $5.760145 \times 10^{-5}$ at time point 9, and finally remains relatively flat. And it can be seen from Figure 6 that when the emergency diesel generator set fails, the probability of system failure is the largest; the system failure rate due to the failure of the auxiliary external power supply, the main external power supply, and the main turbine generator decreases in order. This order is exactly opposite to the order of power supply priority of the electrical system power supply of the nuclear power plant, i.e., the higher the power supply priority, the lower the probability of its failure leading to the failure of the electrical system.

*5.4. Analysis of the Impact of Common Cause Failure Factors on the Reliability of Electrical Systems*

Aiming at the characteristics of each component in the electrical system of a nuclear power plant, a group of common fault failure components is selected for system common fault failure analysis.

(1) Assuming that no common fault failure occurs at startup of auxiliary transformers 1 and 2, the runtime common cause failure is modeled using the $\beta$-factor model $\beta = 0.1$. Then, the common fault failure rate at operation is $c = \lambda \beta_1 = 0.2776 \times 10^{-3}$, and the common fault failure probability of the auxiliary transformer is calculated by applying Equation (4) as $C_1(t) = 2.664936 \times 10^{-0.5} - 2.66493936 \times 10^{-0.5} \times e^{-85.465477}$

(2) Nuclear power units are equipped with emergency diesel generators, which are called EMP and EMQ. Assuming that the probability of startup failure of EMP and EMQ is $\gamma = 0.0236$ and the common fault failure is modeled using a $\beta$-factor model with $B = 0.05$ at startup and $A = 0.1$ at operation, then the probability of initial success of the standby diesel generator EMP and EMQ after considering startup failure is $1 - \gamma = 0.9764$ Then, their initial common fault failure probability is $\gamma_c = \gamma \beta_0 = 0.00118$, the common fault failure rate at operation is $c = \lambda_{LGP,LGQ} \beta_1 = 17.4324$, and the common fault failure probability of the diesel generator is calculated by applying Equation (4) as

$$C_2(t) = 0.00495 - 0.00377 e^{-3521.4324t}$$

(3)  Applying the GO-FLOW method to first calculate the failure probability of the system that does not contain the common fault failure, and then calculate the failure probability of the system that contains the common fault failure according to Equation (4), the total unavailability change curve of the system is obtained, as shown in Figure 7.

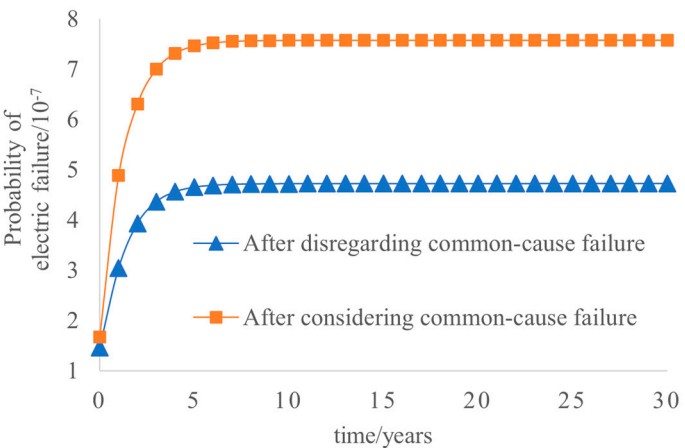

**Figure 7.** Variation curve of total unavailability of electrical system.

As can be seen from Figure 7, the outage probability without considering co-causal failure increases from $1.4562 \times 10^{-7}$ at time point 0 to $4.720932 \times 10^{-7}$ at time point 13, and is finally maintained a relatively smooth state. The outage probability considering common fault failure increases from $1.669504 \times 10^{-7}$ at time point 0 to $7.575017 \times 10^{-7}$ at time point 14, and is finally maintained at a relatively smooth state. The group of common fault failure components considered has a greater impact on the overall reliability analysis of the electrical system of a nuclear power plant. Therefore, in practical engineering applications, especially for redundant systems, the common fault failure factors should be fully considered.

### 5.5. Analysis of the Impact of Additional Standby Units on the Reliability of the Electrical System

If a common fault failure occurs in the standby diesel generator sets, it will have a serious impact on the reliability of the electrical system of the nuclear power plant [25]. To further improve the ability to secure the internal power supply of the nuclear power plant, the nuclear power plant can install additional diesel generators. The probability of power failure of the nuclear power plant can be calculated within 24 h after the failure of the external power source, as shown in Figure 8.

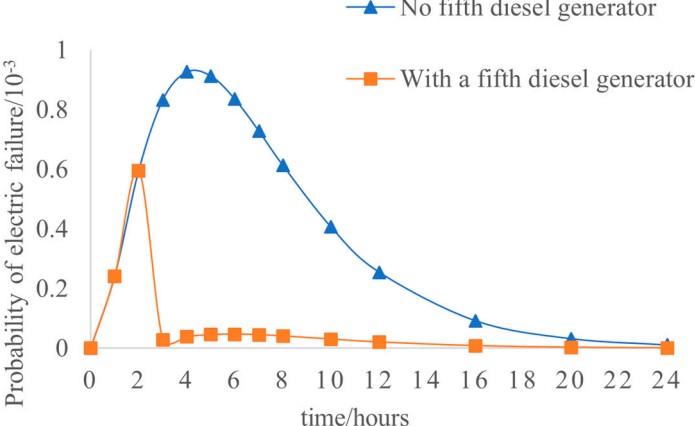

**Figure 8.** Impact of additional diesel generators on the probability of plant-wide power outages.

The emergency diesel generator sets EMP and EMQ may experience a common fault failure. From Figure 8, the power failure probability without the addition of the fifth diesel generator first increases rapidly from 0 at time point 0 to $0.926297 \times 10^{-3}$ at time point 4, and then slowly decreases to 0 at time point 24. The power failure probability with the addition of the fifth diesel generator first increases rapidly from 0 at time point 0 to $5.96036 \times 10^{-4}$ at time point 2, steeply decreases to $0.140665 \times 10^{-4}$, slowly increases to $0.468085 \times 10^{-4}$ at time point 6, and finally, slowly decreases to 0 at time point 24. And from Figure 8, if there is no additional diesel generator and only EMP and EMQ are on standby, the probability of plant-wide power outage of the nuclear power plant reaches its maximum at about 4 h due to the failure of the generator itself, and with the increase in the operation time, the probability of power outage decreases due to the repair of the external power source. With the increase in operation time, the probability of power failure decreases due to the repair of the external power source. The commissioning of additional diesel generators can greatly reduce the probability of power failure. Before the normal operation of the additional diesel generator, the probability of plant-wide power failure reaches the maximum in 2 h, while the probability of power failure decreases abruptly after 2 h.

## 6. Conclusions

For the electrical system of nuclear power plants, a reliability analysis method based on the GO-FLOW method considering the common fault failure is proposed, and the analysis of the arithmetic example verifies the validity of the method. The following conclusions are obtained.

(1) The constructed GO-FLOW model of the external power supply system, auxiliary power supply system, and standby power supply system of a nuclear power plant considers the multi-mode repairable component reliability parameter equivalence model and the improved quantitative calculation method of the common signaling system, which improves the accuracy of the reliability analysis of the electrical system of a nuclear power plant.

(2) The group of common fault failure components has a greater impact on the overall reliability analysis of a nuclear power plant's electrical system, and for redundant systems, common fault failure factors should be fully considered.

(3) The addition of standby diesel generators can greatly reduce the probability of power outages and can effectively improve the reliability of the electrical system in nuclear power plants.

The next step will be to apply relevant measures, such as improving the reliability of the power supply system of nuclear power plants to several nuclear power units, thus further improving the reliability of nuclear power plant production.

**Author Contributions:** Methodology, Z.W.; Software, Y.S.; Writing – review & Validation, J.Z.; Formal analysis, X.D.; Resources, B.W.; Data curation, C.C.; Editing, H.W. All authors have read and agreed to the published version of the manuscript.

**Funding:** This research received no external funding.

**Institutional Review Board Statement:** Not applicable.

**Informed Consent Statement:** Not applicable.

**Data Availability Statement:** No new data were created during the study period.

**Conflicts of Interest:** The authors declare no conflict of interest.

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
