# Peer review of "Reliability Analysis of Nuclear Power Plant Electrical System Considering Common Cause Failure Based on GO-FLOW"

_sustainability, doi:10.3390/su151914071_

Round 1

Reviewer 1 Report

English language usage must be revised throughout the manuscript

Long sentences in abstract and throughout the manuscript must be shortened and corrected

Give details about the nuclear plant investigated in this study

What are effects if power fails? this must be included

What is meant by GOFLOW method? explain briefly

Check the heading number format thoroughly

What is meant by PSA in line 40, Pg. 1?

Give separate nomenclature section. Always use full form during first time use

Citations styles are not uniform. Check them with the manuscript guidelines and follow accordingly

List the objectives of the work as bullet points at the end of introduction section

Novelty of the work also must be reported

he or the, check line 79, 80 Pg. 2

What is mean by UHV, line 86, pg 2.

Nomenclture is missing. must add it

Fig 2 is not clear. Redraw and use slightly large fonts for visibility

Reference is needed for Eqn 1 to 4

Fig 4 needs correction. Check and redraw it

Give reference for data in Table 1, Table 2

What is meant by fifith diesel generator in Fig 8?

Scope for future work must be included in conclusion 

Many typo mistakes

Long sentences

Sentence formation error

Author Response

  1. There was a problem with the English grammatical presentation of lines 272 and 410, which has been corrected.
  2. A long sentence shortened by 14 lines.
  3. Background information on nuclear power plants and systems is provided at 141.The total installed capacity of nuclear power plants is 6.12 million kilowatts (MW), and large-scale nuclear power plants usually have a single unit capacity of 900 MW or 1,300 MW, a transmission level of 400 kV or 500 kV, and megawatt-class pressurized water re-actors.
  4. The introduction was improved in line 29 by emphasizing the importance of reliable electrical systems in the safety of nuclear power, citing the significant impact of the examples of the nuclear accidents at Three Mile Island in the United States in 1979, at Chernobyl in the former Soviet Union in 1986, and at Fukushima in Japan in 2011. In the event of a power outage, the consequences of the above examples would result.
  5. The GO-FLOW method is described on line 72 and details the specific steps and components. GO-FLOW method is a new probabilistic risk evaluation method. It uses a graphical rendition of success as a guide to the system and can assess the reliability and availability of the system. The specific process of the GO-FLOW method for system reliability analysis is as follows:(1) Analyze the given system, determine the scope, function, and included components of the system, specify the system's reliability indexes, and give an engineering or schematic diagram of the system.(2) Determine the inputs and outputs of the system.(3) Determine the minimum set of output signals required for normal operation of the system.(4) Identify the system's reliability indicators. (3) Specify the normal operating state of the system and determine the minimum set of output signals required for normal operation of the system. (4) Create a GO-FLOW diagram and determine the type of GO-FLOW operator to be used according to the function of the unit; the operator should contain all the major units in the system. (5) Define a finite number of discrete time points. (6) Once the GO-FLOW diagram is established, determine the state probability data for all units and then enter the data by operator number. (7) Starting from the input operators, the output signals of the system are computed step-by-step up to the output signals of the system based on the GO-FLOW diagram and the data according to the rules of operation of the operators. (8) Based on the results of the GO-FLOW operation and the system success criteria used to calculate the reliability or availability of the system, the system is evaluated according to the functions and requirements of the system.
  6. Thoroughly checked the heading number format.
  7. Explained the meaning of the abbreviation "PSA" in line 92, “probabilistic safety assessment for nuclear power plants”.
  8. Explained the meaning of the abbreviation "EHP" in line 146, “extra high pressure”.

The meaning of "LOOP events" and "SBO events" is explained in detail between lines 162 and 180. The “LOOP event” :In the GO-FLOW diagram of the external power supply system, signal flow 54 represents the entire external power supply system, and signal flows 34 and 53 represent the outputs of the 2-way power supply of the external power supply system, respectively, and the logical relationship of the 2-way power supply outputs is indicated by the or gate operator 54, and when there is no power supply in the 2-way for the loss of the external power supply system, i.e. The “SBO event”: When the external power system of the nuclear power plant is lost, the external power system and the diesel generator relate to the contingency gate operators 55 and 59 in the diagram. Signal flow 58 and 62 respectively represent the backup system 2-way power supply output, with or gate operator 63 represents the 2-way power supply output of the logical relationship, the output signal 63 failure means that the 2-way are no power sup-ply output, on behalf of the nuclear power plant plant-wide power outage, i.e.

Explaining the meaning of the α-factor and β-factor factors between lines 221 and 226.,For the component group that does not consider maintenance, the probability of common fault failure is calculated by the common fault failure parameter model, and there are existing common fault failure parameter models: α-factor model, β-factor model, and so on. Given the estimated values of the model parameters, it is possible to calculate the probability of common fault failure for a group of components without considering maintenance.

The Markov process is explained in lines 240 through 242,For the above case, a Markov process can be used. Markov process: also called Markov chain, it is a memoryless stochastic process, it can be represented by a tuple <S, P>, where S is a finite number of state sets and P is the state transfer probability matrix.

Explained the meaning of "EMP" and "EMQ" in line 372, Nuclear power units are equipped with emergency diesel generators, which called EMP and EMQ.

  1. Replace all superscripts of cited references with normal superscripts.
  2. Work objectives are set out in bullet point form on line 136. The main objectives of this paper's work are (1) to establish a reliability model of nu-clear power plant electrical systems based on the GO-FLOW method, (2) to study the analysis method of repairable system common fault failure based on the GO-FLOW method, (3) to analyze the process of reliability analysis of nuclear power plant electrical systems based on the GO-FLOW method taking into account the common fault failure, and (4) to carry out the arithmetic case analysis so as to draw the conclusions.
  3. The novelty of this paper is emphasized in line 117. In the current research, fewer scholars have used the GO-FLOW method to analyze the impact of common cause failure factors on system reliability. In this paper, combining with the characteristics of repairable nuclear power plant power supply system, a general method of using GO-FLOW method to analyze the influence of common cause failure fac-tors on the reliability of repairable system is studied.
  4. In line 136, replace "he" with "the".
  5. Explained the meaning of the abbreviation "EHP" in line 146, “extra high pressure”.
  6. Adjusted the font size of Figure 2.
  7. The formula for the system failure rate is derived by citing the formula derivation process in reference [22].
  8. Change "BEGAN" to "START" in the flowchart of the 269-line program. And adjusted text size.
  9. The data in Tables 1 and 2 are quoted from reference [17].
  10. Explain in line 404 that the "fifth diesel generator" refers to the additional diesel generator.
  11. Added future work direction on line 449. The next step will be to apply relevant measures such as improving the reliability of the power supply system of nuclear power plants to several nuclear power units, thus further improving the reliability of nuclear power plant production.

Reviewer 2 Report

In this article, the authors proposed a reliability analysis of NPP electrical systems based on GO-FLOW method. The article is interesting, meaningful, and address an important topic related to the safe operation of NPPs. Some comments are given below:

1- Please consider the appropriate citing style of the sustainability journal. Don't use superscript to cite the references.

2- The first part, introduction is numbered as "0". It should be "1".

3- Moderate English language corrections are required.

4- There are many unresolved abbreviations scattered throughout the whole text.

5- Start the flow chart shown in Fig.4 with "start".

6- Figures 5-8 are unclear "blurry".

7- Referral to figures, sometimes you use: "Fig." and "Figure". Single style must be used.

Moderate English language editing is required.

Author Response

  1. Replace all superscripts of cited references with normal superscripts.
  2. Renumber the introductory section from "0" to "1" and add one to all subsequent headings accordingly.
  3. There was a problem with the English grammatical presentation of lines 272 and 410, which has been corrected.
  4. Explained the meaning of the abbreviation "PSA" in line 92, “probabilistic safety assessment for nuclear power plants”.

Explained the meaning of the abbreviation "EHP" in line 146, “extra high pressure”.

The meaning of "LOOP events" and "SBO events" is explained in detail between lines 162 and 180. The “LOOP event” :In the GO-FLOW diagram of the external power supply system, signal flow 54 represents the entire external power supply system, and signal flows 34 and 53 represent the outputs of the 2-way power supply of the external power supply system, respectively, and the logical relationship of the 2-way power supply outputs is indicated by the or gate operator 54, and when there is no power supply in the 2-way for the loss of the external power supply system, i.e. The “SBO event”: When the external power system of the nuclear power plant is lost, the external power system and the diesel generator relate to the contingency gate operators 55 and 59 in the diagram. Signal flow 58 and 62 respectively represent the backup system 2-way power supply output, with or gate operator 63 represents the 2-way power supply output of the logical relationship, the output signal 63 failure means that the 2-way are no power sup-ply output, on behalf of the nuclear power plant plant-wide power outage, i.e.

Explaining the meaning of the α-factor and β-factor factors between lines 221 and 226.,For the component group that does not consider maintenance, the probability of common fault failure is calculated by the common fault failure parameter model, and there are existing common fault failure parameter models: α-factor model, β-factor model, and so on. Given the estimated values of the model parameters, it is possible to calculate the probability of common fault failure for a group of components without considering maintenance.

The Markov process is explained in lines 240 through 242,For the above case, a Markov process can be used. Markov process: also called Markov chain, it is a memoryless stochastic process, it can be represented by a tuple <S, P>, where S is a finite number of state sets and P is the state transfer probability matrix.

Explained the meaning of "EMP" and "EMQ" in line 372, Nuclear power units are equipped with emergency diesel generators, which called EMP and EMQ.

  1. Change "BEGAN" to "START" in the flowchart of the 269-line program.
  2. The curves in figures 5 to 8 have been adjusted to be finer and the punctuation more visible, while the legend layout has been modified to make the pictures clearer.
  3. Amend all "Fig." in the article to read "Figure".

Reviewer 3 Report

I have one major comment:

1. The analysis part of this paper should be deeper with more insight views or quantitative result presented in figures.

Author Response

  1. Specific descriptions of the figure data (maximum values, minimum values, time points of trends) are added to each of Figures 5-8, and the trend of each plot over time is described.

Reviewer 4 Report

1) The introduction can be further refined to explicitly emphasize the pivotal importance of reliable electrical systems in nuclear power plants. Clearly outline the context of common cause failure in such systems and its potential implications for safety.

2) Expand on the concept of common cause failure, highlighting its intricate interplay within the nuclear power plant context. Offer more detailed examples or scenarios to showcase its significance.

3) Elaborate on the GO-FLOW method, detailing its specific steps and components. This would provide readers with a clearer understanding of how the proposed approach is constructed and how it can address the challenges of common cause failure.

4) Since reliability analysis often involves complex mathematical models, consider providing supplementary explanations or derivations of the equations used. This will assist readers in comprehending the technical aspects of the proposed methodology.

5)  Consider incorporating a brief overview of existing reliability analysis methods for nuclear power plant electrical systems. This could help readers better grasp the unique contributions and advantages of the GO-FLOW method.

6) Offer more contextual information about the nuclear power plants and systems under consideration. This can include details about the scale, type, and operational conditions of these plants, which would enrich the practical relevance of the study.

Author Response

  1. The introduction was improved in line 29 by emphasizing the importance of reliable electrical systems in the safety of nuclear power, citing the significant impact of the examples of the nuclear accidents at Three Mile Island in the United States in 1979, at Chernobyl in the former Soviet Union in 1986, and at Fukushima in Japan in 2011.
  2. Expansion of the concept of common fault failure in line 108, listing specific causes and emphasizing its complex effects in nuclear power plants. Common Fault Failure (CFF) is the simultaneous failure of several units in a system due to some common reasons (e.g., changes in the environment, events such as earthquakes or lightning strikes, operational errors in maintenance, other human interference, etc.). Since the unit normal or fault states are statistically related to each other in a system with common fault failure, the reliability data processing and analyzing methods of the system are quite different from the traditional ones, which brings more difficulties to the reliability analysis of the system. Probabilistic risk analysis in the nuclear industry shows that common fault failure is one of the main reasons for system failure, equipment unavailability, and risk in nuclear power plants.
  3. The GO-FLOW method is described on line 72 and details the specific steps and components. GO-FLOW method is a new probabilistic risk evaluation method. It uses a graphical rendition of success as a guide to the system and can assess the reliability and availability of the system. The specific process of the GO-FLOW method for system reliability analysis is as follows:(1) Analyze the given system, determine the scope, function, and included components of the system, specify the system's reliability indexes, and give an engineering or schematic diagram of the system.(2) Determine the inputs and outputs of the system.(3) Determine the minimum set of output signals required for normal operation of the system.(4) Identify the system's reliability indicators. (3) Specify the normal operating state of the system and determine the minimum set of output signals required for normal operation of the system. (4) Create a GO-FLOW diagram and determine the type of GO-FLOW operator to be used according to the function of the unit; the operator should contain all the major units in the system. (5) Define a finite number of discrete time points. (6) Once the GO-FLOW diagram is established, determine the state probability data for all units and then enter the data by operator number. (7) Starting from the input operators, the output signals of the system are computed step-by-step up to the output signals of the system based on the GO-FLOW diagram and the data according to the rules of operation of the operators. (8) Based on the results of the GO-FLOW operation and the system success criteria used to calculate the reliability or availability of the system, the system is evaluated according to the functions and requirements of the system.
  4. The formula for the system failure rate is derived by citing the formula derivation process in reference [22].
  5. A brief description of existing reliability analysis methods for electrical systems in nuclear power plants is given in line 47. At present, there are mainly these commonly used methods for reliability analysis of nuclear power plant electrical systems: (1) Fault tree analysis method, which consists of specific logic gates and various fault events, has a simple structure and is easy to analyze. However, the modeling is greatly influenced by human factors, and it is easy to miss or repeat. (2) Monte Carlo method, which is easy to model complex systems and the model can take time constraints, parameter distributions, etc. into account. However, it requires a lot of operations, which is expensive and time-consuming. (3) Markov state transfer method, which can explicitly consider the effects of changes in process variables on the system state and the system control rules, as well as the effects of time changes on the system state. However, for large and complex systems, as the number of components in the system increases, the states in the Markov model will increase exponentially, and it is more difficult to use the Markov state transfer model alone for dynamic probabilistic evaluation. (4) GO analysis method, which can directly simulate the interactions and correlations of the system and components, and directly perform the system success probability analysis in a success-oriented manner. However, there are more types of operators and the use of them is complicated, which requires analysts not only to be very familiar with the system but also to have an in-depth understanding of the GO method. In addition, the GO method program development is more complicated, and there is no complete calculation program yet. (5) GO-FLOW analysis method, easy to model, change and verify. GO-FLOW diagram can fully express the relationship and interaction between components and systems and can well describe and reflect the system structure.
  6. Background information on nuclear power plants and systems is provided at 141.The total installed capacity of nuclear power plants is 6.12 million kilowatts (MW), and large-scale nuclear power plants usually have a single unit capacity of 900 MW or 1,300 MW, a transmission level of 400 kV or 500 kV, and megawatt-class pressurized water re-actors.

Round 2

Reviewer 1 Report

My Comments have been addressed

Author Response

Optimization adjustments have been made accordingly.

Reviewer 3 Report

Two comments:

1. Please make sure the data listed in Table 1 are reasonable.

2. Please provide more deep insightful view by adding figures to improve the scientific soundness of this paper.

The language of this paper needs some improvement to become more authentic. 

Author Response

  1. All data in Table 1 are from reference [17].
  2. A table of reliability data on the repairable components of this nuclear power unit from reference [17] has been added in line 331 to show its scientific nature more profoundly.

Reviewer 4 Report

Paper can be accepted in the present form.

Author Response

The English grammar is appropriately adjusted and a table of reliability data on the repairable components of this nuclear power unit from reference [17] is added in line 331 to show its scientific nature more profoundly.

Round 3

Reviewer 3 Report

Issue addressed. I recommend this paper for publication.